# Influence of the Temperature-Strain Parameters on the Structure Evolution and Carbide Transformations of Cr-Ni-Ti Stainless Steel

**DOI:** 10.3390/ma15082784

**Published:** 2022-04-11

**Authors:** Andrei Rudskoi, Georgii Kodzhaspirov

**Affiliations:** Materials Science Centre, Peter the Great Saint-Petersburg Polytechnic University, 195251 St. Petersburg, Russia; rector@spbstu.ru

**Keywords:** thermomechanical processing, austenitic stainless steel, microstructure, dynamic recrystallization (DRX), carbide transformation

## Abstract

Influence of strain accumulation schedule during hot rolling, under the thermomechanical control process (TMCP) on the structure evolution and carbide transformations of Cr-Ni-Ti austenitic stainless steel, is studied. The cellular, fragmented dislocation substructure and dynamically recrystallized (DRX) structure are observed in the steel with different strain accumulation schedules. It was found that the strain accumulation schedule, especially fractionality, affects the work-hardening and softening behavior quite significantly. The role of the strain accumulation schedule on the fragmented substructure and DRX structure evolution as well as carbide transformations and the relationship between the microstructure changes due to TMCP and the mechanical properties of studied steel, involving the recent ideas of the physics of large plastic strains, are considered.

## 1. Introduction

Austenitic stainless steels have been widely used in a variety of industrial fields due to their superior corrosion resistance and excellent mechanical properties. However, the relatively low yield strength of austenitic stainless steels is an obstacle to wide-spread applications. As well known, TMCP is one of the most advanced resource-saving technologies for the manufacture of metal blanks and parts [1,2,3]. A recent report found that TMCP can make the grain in austenitic stainless steels more refined as a result of dynamic recrystallization (DRX) and substructure strengthening, which increase strength without significant reduction in ductility and toughness [3,4,5]. The application of TMCP to the most widely used (AISI 321 type) austenitic stainless steels could produce high-strength plates and stocks with yield strength of higher than 400 [6,7] and 600 MPa for high nitrogen steels [7,8]. Based on the hot rolling process, TCMP is a complex superposition of fine structure, subgrain/grain evolution, and phase changes due to transformation, particle dissolution, and precipitation. Refining of austenite grains and subgrains is possible by utilizing recrystallization and fragmentation under specific thermomechanical conditions. The structure formation over the whole cycle of TMCP is sensitive to small additions of carbide forming elements, such as Ti, Nb, and V, which hinder subboundary and boundary migration. Therefore, this affects the formation of the final sub- and grain structure, which is responsible for the mechanical and functional properties [9].

The role of DRX and some examples of the investigations dealing with the study of the evolution of grain boundaries and subboundaries in stainless steel during DRX have been described in [10,11,12,13,14,15,16,17]. Several investigations involving the study of transmission electron microscopy (TEM) applications of dislocation structure, which are formed during hot rolling at different deformation temperatures and strain degrees in one-pass reduction, are described in [18]. Under normal conditions, hot rolling is usually performed by several passes. Although the structure formation mechanism and the effect of small additions of carbide forming elements on the final sub- and grain structure of several steel grades have been recognized and extensively studied, an investigation of the comparative effect of fractionality hot deformation with one-time and gradual strain accumulation on the structure, carbide transformations, and mechanical properties of this type of steel has not been comprehensively studied. As a result, this research aims to explore the governing mechanism of structure formation, including the evolution of dislocation substructures, subgrain misorientations, and carbide transformations of Cr-Ni-Ti stainless steel during hot rolling with one-time and fractional strain accumulation. Furthermore, particular attention is focused on the quantitative assessment of the relationship of dislocation substructure and carbide transformations, taking place under the studied thermomechanical regimes of TMCP with mechanical properties.

## 2. Materials and Methods

Under normal conditions, hot rolling is usually performed by several passes. At different ranges of strain accumulation schedules, the corrosion-resistant austenitic steel (08X18N10T-type AISI 321, Electrostal, Russia and Cleveland, U.S.A.), of type Cr-Ni-Ti (0.06% C; 17,5% Cr; 10% Ni; 03% Ti) has been studied. Specimens with a cross-section of 22 mm × 22 mm were heated for 40 min at 1180 °C. Then, they were air-cooled to the studied deformation temperature and rolled on a laboratory rolling mill (DUO 210, Moscow, Russia) at 0.35 m/s. In the case of one-pass, the strain degree during rolling was 

12,31,51%, and during fractional rolling—10% per pass. One-time deformation may be applied with strain degrees of 31 and 51%. In the third (1060 °C) and fifth (1010 °C) passes, rolling was carried out after cooling from a heating temperature of 1180 °C to temperatures corresponding to the deformation temperature during the implementation of the fractional mode, respectively. Metallographic analysis has been performed using the light microscope Zeiss Axio Observer Dlm (Jena, Germany) with an image analysis system. The fraction of the recrystallized structure (recrystallized grains, Δ^r^, %) along the section of the workpiece for all of the studied modes was estimated using the methods of stereometric analysis and computer metallography [19]. To study the dislocation substructure, samples from blanks processed according to various modes were examined using a JEM-200 CX transmission electron microscope (Tokyo, Japan). The mechanical properties of steel were determined on samples cut from rolled workpieces along the rolling direction, according to ISO6892-84 specifications. Moreover, the tests were carried out on a Zwick//Roell Z050 universal testing machine(Ulm, Germany).

## 3. Results

### 3.1. Microstructure

In the initial state (40 min at 1180 °C followed by water cooling), the microstructure of steel, which is characterized by large grains containing annealing twins and round carbide inclusions of a few microns, was uniformly distributed throughout the sample volume. Under conditions of fractional reduction, an increase in the total degree of reduction results in the monotonic diminution of grain size and growth of precipitate density ρ^0^.

In addition, the length of the grains in the rolling growth direction noticeably increases, similar to Σ*ε_i_*. A rise in the strain degree of one-pass deformation has a completely different effect on the initial structure. Both the grain size and the precipitate density change non-monotonically. The grain sizes are smallest at *ε* = 30% (see Table 1), while the precipitate quantity is largest, although it does not significantly grow in any case. A further rise in the strain degree of one-pass reduction to 50% causes a slight growth in $\bar{D}$ and a small decrease in ρ^0^ (Table 1). Investigations of fine structure using transmission electron microscopy (TEM) show that in the initial state the dislocation density of the steel is low, at about 10^8^ cm^−2^. The boundaries are perfect and rectilinear, and do not contain precipitates. Both fractional and one-pass hot plastic deformation radically alter the initial structure of the steel. This is evidenced in the fact that as the dislocation density increases, new misorientation boundaries of different forms appear, the structures of existing boundaries change, and finely dispersed carbide precipitates appear.

#### 3.1.1. Fractional Deformation

As the number of passes increases, a monotonic growth of dislocation density ρ^d^ takes place, which changes the most from 10^8^ to 1.5 × 10^10^ cm^−2^ after the first pass. The subsequent deformation only has a slight influence on ρ^d^, which increases to 2.0 × 10^10^ cm^−2^ after three passes, and 2.3 × 10^10^ cm^−2^ after five passes (see Table 1). The spatial distribution of dislocations also changes. As *ε* = $\sum \varepsilon_{i}$ grows, the characteristic indications of dynamic recovery are more clearly seen in the dislocation structure. At first, a weakly expressed cellular structure with no greater than 0.1° of misorientation is formed (Figure 1a). Then, individual boundaries elongated in the rolling direction appear against that background, causing misorientations of about *θ* = 1…3 deg. The distances between them change from a few to tens of microns. The places where the boundaries break down, lines of partial disclinations, are sources of strong internal stresses [20,21]. After rolling with *ε* = 10%, these boundaries are found in all of the grains. However, they form regions of fragmented structure in the most favorable oriented grains (Figure 1b). The fragments are microregions of ~0.5 μm in size, enclosed on all sides by dislocation boundaries and misoriented relative to the surrounding matrix or neighboring fragments by angles of 1…3 deg. The volume fraction occupied by a fragmented structure is only 5–10% after the first pass. As the deformation accumulates, a larger volume of material becomes fragmented. Therefore, after the third pass Δ^f^ reaches 60–70%, and after the fifth it reaches 80–90%. The fragmented structure becomes more perfect, the quantity of torn-off boundaries diminishes, and their dislocation structure becomes ordered. The fragments become finer on average, and the misorientation between them increases appreciably. In Figure 1b,c, the numbers 1–6 denote fragments on which the misorientation vectors *θ*_gs_ were determined with the techniques [18]. The results are given in Table 2. After one-pass reduction, the misorientation angles are 1…3 deg. They increase sharply as a result of the five-pass reduction, reaching several tens of degrees on individual boundaries: *θ*_1,4_ = 20.3 deg., *θ*_2,4_ = 12.2 deg., etc. Another peculiarity in the evolution of the fragmented structure is that the fragments grow progressively longer in the rolling direction. Moreover, there is a substantial change in the structure of grain boundaries during deformation.

Furthermore, a large number of dislocations and finely dispersed precipitates appear on the fragments, they are no longer rectilinear, and become facetted. Partial disclinations are emitted inside the boundaries with the deformation step. As a result, on average, fragmentation becomes more intensive in the boundary zones than in the volume. Another distinctive feature of the influence of fractional deformation on the structure of steel is the appearance of dispersed precipitates inside the grains, which is absent from the initial material. The volume density of these particles increases and their size decreases as the number of passes increases. After 10% of one-pass deformation, only individual precipitates are observed. After three passes, ρ^e^ ≈ 3 × 10^12^ cm^−3^, and after five passes, ~2 × 10^14^ cm^−3^. The mean size of the particles drops from 0.1 … 0.3 μm to ~0.01 μm. The phase composition of the precipitates was determined by an analysis with the single reflection technique [22], which turned out to be titanium carbide (TiC). Precipitation of TiC particles is due to plastic deformation and apparently involves the following mechanism. Hot plastic deformation results in partial dissolution of large precipitates of the initial carbide phase and enrichment of the solid by the carbon, which was already bound in these precipitates. When decelerating deformation or cooling, the supersaturated solid decomposes and dispersed titanium carbide is precipitated uniformly throughout the volume. Of note, no traces of recrystallization were found in the deformed samples, according to fractional deformation modes corresponding to fractional deformation.

#### 3.1.2. One-Pass Deformation

The evolution of the structure differs qualitatively from the aforementioned structure, as the strain degree of one-pass reduction increases. The main difference consists in the appearance of DRX regions, which is observed starting from *ε* = 30%. At first, they are practically dislocation-free regions of regular ellipsoidal shape, that are not larger than a few microns in size and with perfect high-angle boundaries (Figure 1d and Figure 2a).

With the development of recrystallization processes, the size of these areas grow and become comparable, and in some cases they exceed the size in the initial state. Of note, there is a high degree of heterogeneity in the size of recrystallized regions, which can vary from ~1 μm (Figure 2a) to ~20–30 μm. At the same time, however, the mean grain size is appreciably smaller than in the initial state. The dislocation density in a recrystallized region can vary within wide limits, depending on the stage at which it appeared, from almost the total absence of any dislocations to ρ^d^ ≈ 10^10^ cm^−2^. However, the spatial distribution of dislocation inside the recrystallized regions always remains uniform: The dislocations do not form braids or tangles, nor a cellular, and even less, a fragmented structure. Annealing twins are frequently seen in the large recrystallized regions, but they are significantly smaller than in the initial or fractionally-deformed state and do not normally exceed a few microns in size (Figure 2b). The boundaries of these twins do not show signs of deformation faceting. Moreover, no release of the carbide phase appears. The proportion of recrystallized structure in the volume of steel monotonically increases with the rising strain degree, reaching 90–95% with a strain degree of 50% (Table 1).

The dislocation structure changes differently in grains not subjected to DRX. The intensity of fragmentation increases with the strain degree, as we can see from the diminution in size to ≤1 μm after *ε* = 50% (Figure 2c) and the growth of misorientation between them to 5 … 10 deg. It can be seen from the distinct stripe contrast of many of the boundaries of fragments and the absence of torn-off boundaries that they are perfect. There are many fewer individual dislocations inside the fragments than during fractional rolling to the same strain degree, and the dislocation density drops with the rise in the strain degree. Fragmentation normally embraces the whole grain. The proportion of fragmented structure in the steel grows from 5…10 to 20…30% as the one-pass reduction grows from 10 to 30%, and then drops to the same 5…10%, thus differing markedly from fragmentation during fractional deformation.

### 3.2. Carbide Transformations

The character of the carbide transformations also differs fundamentally: As the strain degree of the one-pass reduction increases, the density of finely dispersed precipitates in the steel drops. Therefore, after *ε* = 30%, only individual dispersed TiC precipitates are seen, which is similar in shape and size to those present in the steel after 10% reduction. They have practically disappeared from the structure of the steel after 50% reduction.

## 4. Discussion

In this study, the changes in the defect structure of the steel result from three competing processes, which allow the entry of the other: Dislocation work-hardening, dynamic recovery, and dynamic recrystallization. The first process is predominant at the initial state of deformation. As *ε* grows, the density of uniformly distributed dislocations grows until the mean interaction forces approach, which operate on the dislocations from external stresses. Collective forms of motion appear in the dislocation ensemble, resulting in substantial rearrangement of the structure: Division of the grain interior first into weakly misoriented cells and then into fragments. On the phenomenological level, these processes can be associated with dynamic recovery. According to contemporary ideas on the plastic deformation of crystals, rotational plasticity modes have appeared in the crystal [20]. A uniformly deformable material cannot further dissipate the mechanical energy supplied to it at a given load rate only by means of plastic shears. Therefore, it divides into a set of misoriented microregions (cells, fragments), each of which starts to swing plastically during the deformation, thereby absorbing additional portions of mechanical energy. As the load rate increases, which in our case is equivalent to an increase in the strain degree of one-pass reduction, the rotational modes and their structural indication—fragmentation—will continue to intensify. This continues until the rate of mechanical energy supplied to the specimen exceeds the threshold value, at which a fragmented structure becomes unstable. As soon as that occurs, dynamic recrystallization develops for a single-phase material, the last and most powerful structural mechanism of energy dissipation. With these general ideas in mind, it is simple to explain the pattern of structural transformations, which we have observed in these experiments. The load rate for one-pass reduction of 10% corresponds to the stage of intensive dynamic recovery which, as it proceeds during each pass of fractional deformation, results in continuous intensification of the fragmentation effect [7,20,21]. This is expressed most clearly in the continuous increase in the angles of turn of adjacent fragments which, as these experiments have shown, reach as much as 20 deg. after five-pass deformation. An increase in the rate of energy supplied to values corresponding to 30% reduction and the corresponding intensification of fragmentation lead to an increase in the misorientation angles of large-angle boundaries. The volumes bounded by the latter are natural nuclei of dynamic recrystallization, which can grow by means of boundary migration, according to the Bailey–Hirsch mechanism as energy dissipation increases, resulting in the formation of new grains. For this reason, the recrystallization threshold is lowered as the degree of deformation rises. In fact, dynamic recrystallization significantly develops during the 50% one-pass reduction. In this case, it becomes the dominant mode of structural transformation. Moreover, fragmentation (in those regions which remain unrecrystallized) intensifies as compared with the 10% strain degree, but significantly less than fractional deformation. We can see this by comparing measurements of misorientations in materials, which have experienced 50% strain degree in conditions of one and multi-pass rolling (Table 2). The structural transformations in the observed variations are in good agreement with the mechanical behavior of the steel both during and after rolling. The strength of the material grows from 315 MPa after the first pass, to 415 MPa after the fifth pass. The plasticity parameters of the steel remain at their former level. As the strain degree of one-pass increases, YS first grows slightly to 332 MPa, and then drops suddenly to 246 MPa, which can be attributed to intensive recrystallization.

To explain the influence of the method of accumulating deformation on carbide transformations in the steel at the conditions of the experiment, the following factors must be considered.
The thermodynamic stability of titanium carbide (TiC) is a fairly stable activity, and its particles of different degrees of dispersity are present in the steel up to 1150 °C.The rolling temperature conditions in the given experiment are selected as a function of the accumulation method of deformation. With fractional deformation, the temperature of the onset of rolling for all of the total degrees of reduction *ε*_Σ_ is T^on^ = 1150 °C. In one-time deformation, rolling with strain degree *ε* = 10% also starts at 1150 °C, but rolling with *ε* = 30 and 50% occurs after the cooling samples are preliminarily heated to 1150 °C to temperatures corresponding to the third (1070 °C) and fifth (1020 °C) pass in fractional deformation (this is carried out in order that the temperature at the end of deformation is the same as the total strain degree).In steel deformation, there is a solution of finely dispersed carbide, which is based on their interaction with the flux of moving dislocations. Freely moving carbon atoms enter the solid solution or segregate at dislocations in the form of Cottrell atmospheres. An increase in the magnitude or rate of deformation is accompanied by an increase in the degree of solution of the initial carbide phase.With preliminary cooling of the steel or cooling in the pauses between deformation, the solid solution corresponding to the limit of solubility of C, Ti atoms fractionally become supersaturated, and break down, with the deposition of carbide phase particles. As more dislocations and inter-fragment boundaries are present in the metal volume, the number of sites for carbide particle deposition increases and these particles are found to be more dispersed and homogeneously distributed throughout the volume.

The aforementioned peculiarities have determined the kinetics of structure transformations. Therefore, it may be proposed that the lowering of strengthening tendency should take place when the solubility temperature of carbide decreases.

Mechanical properties.

The laws of structural changes observed in the present work with one-time and fractional accumulation of deformation in Cr-Ni-Ti austenitic stainless steel are of practical importance, since they permit the prediction of its behavior in rolling and mechanical tests. For example, with fractional accumulation of deformation, a monotonous increase in strength and decrease in ductility can be expected, which is associated with an increase in the density of dislocations, dispersed releases of the carbide phase, and the proportion of fragmented structure in the volume (see Table 1). In the case of single-pass deformation, as the strain degree increases from 10 to 30%, an increase in strength is observed, and with a further increase in the strain degree to 50%, a drop in strength and growth in ductility is observed. This behavior is explained by a change in the ratio of the fraction of the recrystallized and fragmented structures. Namely, with an increase in the fraction of the recrystallized structure and a decrease in the fraction of the fragmented structure, there is a drop in strength and growth in ductility (see Table 1).

## 5. Conclusions

The strain accumulation schedule, during hot rolling under TMCP conditions, affects the strengthening and softening behavior of Cr-Ni-Ti austenitic stainless steel quite significantly. The monotonic growth in strength at the fractional accumulation of deformation results in an increase in the density of dislocations and precipitates, as well as a rise in fragmented substructure quantities. The magnification of one-time deformation reductions results in DRX volumes. As more dislocations and inter-fragment boundaries are present in the metal volume, the number of sites for carbide particle deposition increases. Furthermore, these particles are found to be more dispersed and homogeneously distributed throughout the volume.

## Figures and Tables

**Figure 1 materials-15-02784-f001:**
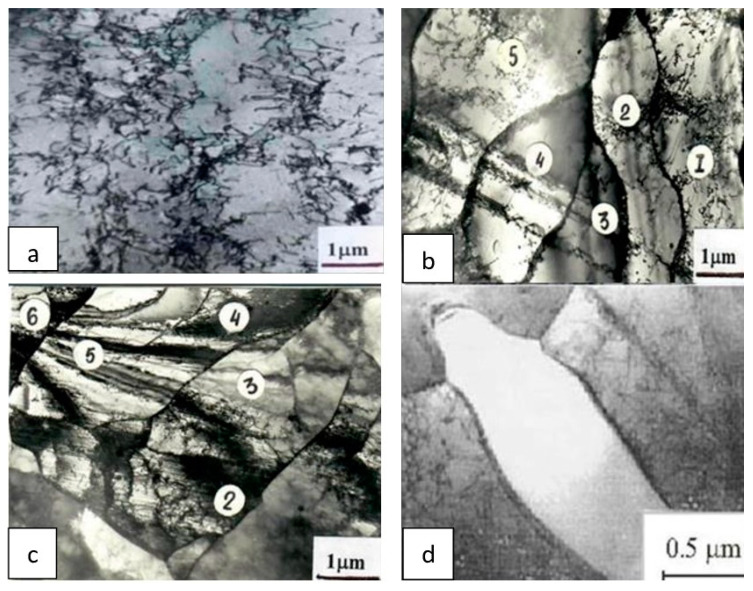
Transmission electron micrographs of weakly misoriented cellular (**a**), fragmented (**b**,**c**), and recrystallized (**d**) structures in Cr-Ni-Ti austenitic stainless steel after fractional deformation (**c**) and after single-stage deformation (**a**,**b**,**d**): (**a**) Number of passes n = 1, *ε* = 10%; (**b**) n = 1, *ε* = 10%; (**c**) n = 5, *ε* = 10%; (**d**) n = 1, *ε* = 30%; 1–6 fragments on which the crystallographic orientation (*θ*) has been determined.

**Figure 2 materials-15-02784-f002:**
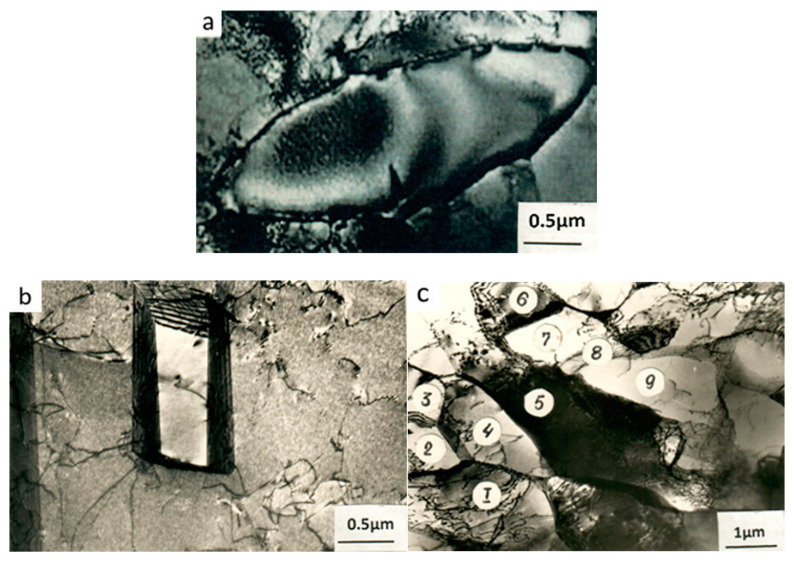
Transmission electron micrographs of Cr-Ni-Ti austenitic stainless steel after one-pass reduction: (**a**) Nucleus of new grain appearing during dynamic recrystallization, n = 1; *ε* = 30%; (**b**) nucleus of annealing twin, n = 1; *ε* = 50%; (**c**) fragmented structure area, n = 1; *ε* = 50%; 1–9: The same, as shown in numbers 1–6 in Figure 1.

**Table 1 materials-15-02784-t001:** Mechanical properties and structure parameters of Cr-Ni-Ti austenitic stainless steel after TMCP with different strain degrees and number of passes.

*n*	*ε*, %	YS, MPa	A, %	$\bar{D}$ μm	ρ^0^, cm^−3^	ρ^d^, cm^−2^	ρ^e^, cm^−3^	Δ^r^, %	Δ^f^, %
1	10	315	52,7	47	11.5 × 10^7^	1.4 × 10^10^	-	0	10
1	30	332	54,5	25	20.5 × 10^7^	1.2 × 10^10^	-	31	31
1	50	246	55,1	31	16.1 × 10^7^	0.5 × 10^10^	-	90	10
3	30	394	50,6	38	14.2 × 10^7^	2.1 × 10^10^	3 × 10^12^	0	61
5	50	415	46,1	16	25.6 × 10^7^	2.4 × 10^10^	2 × 10^14^	0	91

Note. YS: Yield strength; A: Elongation; D: Mean grain size; ρ^0^: Density of precipitates according to optical metallographic results; ρ^d^: Dislocation density; ρ^e^: Density of finely-dispersed precipitates from electron microscopic results; Δ^r^: Share of recrystallized structure; Δ^f^: Share of fragmented structure.

**Table 2 materials-15-02784-t002:** Misorientation in Cr-Ni-Ti austenitic stainless steel after TMCP, according to modes with different numbers of passes (n) and strain degrees (*ε*).

n	*ε*, %	s-s^0^	*θ*, deg.	n	*ε*, %	s-s^0^	*θ*, deg.
1 1 1 1 1 1 1 5 5 5 5 5 5 5	10 10 10 10 10 10 10 50 50 50 50 50 50 50	1–2 1–5 2–3 2–4 2–5 3–4 4–5 1–2 1–4 2–3 2–4 3–4 4–5 5–6	0.7 2.4 1.7 2.5 2.1 0.9 1.8 8.4 20.4 4.6 12.2 8.0 4.4 4.6	1 1 1 1 1 1 1 1 1 1 1 1	50 50 50 50 50 50 50 50 50 50 50 50	1–2 1–4 2–3 2–4 3–4 4–5 5–6 5–7 5–8 5–9 6–7 7–8	1, 4 4, 3 3, 1 5, 4 2, 4 8, 5 1, 7 1, 5 3, 5 4, 7 1, 9 2, 8

## Data Availability

Not applicable.

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
