# Peer review of "Influence of the Temperature-Strain Parameters on the Structure Evolution and Carbide Transformations of Cr-Ni-Ti Stainless Steel"

_materials, 2022, doi:10.3390/ma15082784_

Round 1
Reviewer 1 Report
- The logic in Introduction should be improved. The problems and the creative points should be declared in this section.
- In section 2, can the authors provide some photos to straightforwardly declare the materials and method? It is important.
- In section 3, the subtitles are suggested to clearly illustrate the major points, namely, 3.1, 3.2….
- Please check the Table 2. Some spelling problems exist.
- Some photos on these minerals should be given in the major part.
Author Response
Point 1: In accordance with the Reviewer's recommendation, the logic in the introduction has been improved and problems and creative moments have been emphasized.
Point 2: The steel under study belongs to the widely used class of austenitic corrosion-resistant Cr-Ni steels, which are well described in many works, in this regard, it seems inappropriate to load the manuscript with this information, at the same time the chemical composition of the main elements is presented in the manuscript. As for the methods, the article contains references to all the techniques used in the work (see references 19, 21, 22), and the work itself is devoted to the study of the influence of the mode of accumulation of deformation on the evolution of dislocation structure and carbide transformations, in the analysis of the results of which the techniques described in the above-mentioned works were used.
Point 3: In accordance with the Reviewer's recommendation, the main points are more clearly outlined, namely 3.1; 3.2...
Point 4: Spelling problems has been corrected.

Reviewer 2 Report
Review of the Manuscript:
- Authors have studied the strain accumulation on the structure evaluation and carbide transformations for stainless steel.
- Various physical properties including microstructure, deformation and transformations have been investigated for the same system.
- Overall, interesting results have been obtained and discussed with proper depth and appropriate explanation.
- Manuscript requires further minor revision prior to get considered for publication in this high impact journal.
**** **** ****
- Introduction needs to be expanded based on related literature survey and concern processes reported in last decade or more.
- Introduction should not cover the practices performed for the present study (please remove last sentence from the current introduction section).
- Reference work is too poor. Authors have to provide more citations based on related work, applications, mechanisms and understanding for the similar materials.
- Write the full form of TEM at its first writing position in the manuscript.
**** **** ****
- What is the aim of the study? What are clear objectives of the study (write this at the end of the introduction)?
- How authors have decided studied contents of different elements, C, Cr, Ni and Ti, in their study?
Author Response
Point 1:. In accordance with the Reviewer's recommendation, the introduction has been expanded on the basis of a review of additional literature and, correspondingly, the references to the relevant literature have been supplemented.
Point 2: The full form of TEM (transmission electron microscopy) at its first writing position is presented (line 46) of the corrected manuscript.
Point 3: Сlear objectives of the study are mentioned at the end of the introduction of the corrected manuscript, namely: “... The aim of the present work is to study the effect of hot rolling with one-time and fractional strain accumulation on the evolution of dislocations structure, subgrain misorientations carbide transformations during the studied regimes of TMCP, and to evaluate the relationship between structural and phase transformations and mechanical properties. “
Point 4: The chemical composition of steel was determined by the "wet chemistry" method and the content of elements in the carbide phase was determined by the technique described in the reference [22].

Reviewer 3 Report
This paper presented the microstructural study on an austenitic steel during hot rolling. The effects of rolling parameters (passes) have been studied in terms of the microstructure. the so-called dynamic recovery and dynamic recrystallisation have been identified by the work. It is very interesting to see these results with advanced characterisation techniques. However, the reviewer would suggest that revision is required to improve the manuscript. Below are some suggestions.
- English should be improved significantly, as the reviewer can see a lot of grammar mistakes in the manuscript. It is difficult to understand on some sentences which may cause misunderstanding to the work. For instance, Lines 49, 217 in the manuscript.
- The figures should be re-organised. Currently there are only two figures in total. But the reviewer is not satisfied with the analysis and discussion secions of the manuscript, especially when the figures (microstructures) are discussed. It is suggested that the authors focus on discussing only one point of conclusions with one figure.
- In Table 1, it shows a clear trend for some of the parameters, e.g. the dislocation density. The authors also discussed this in the context. In the case, a plot showing the trend of dislocation density at different conditions may look more interesting, which can also support the findings of this study.
- As the research is investigating the microstructure of the selected material. The initial microstructure should be added for readers to compare the microstructural changes before and after deformation.
- Some of conclusions or statements are not well evidenced in the Discussion chapter. For example, the Mechanical properties (lines 309 - 311) in the chapter is not well supported by any evidence.
Author Response
Response to Reviewer 3 Comments
Point 1:. In accordance with the Reviewer’s remark English corrected, including Lines 49, 217.
Point 2: The analysis and discussion of the sections and all the figures are corrected. Analysis of the Fig.1 presented in the corrected manuscript section 3.1.1 and the analysis of the Fig.2 is presented in 3.1.2.
Point3: In our opinion, experimental currents are not enough for graphical interpretation of changes in dislocation density.
Point 5: The discussion of the correlation of structure and mechanical properties is supplemented in the section: “Mechanical properties” (Lines 323-336).
Point 4: The objective of this work was to study the effect of the deformation accumulation mode on the dislocation structure, carbide transformations and mechanical properties. Comparison with the initial undeformed structure was performed in our earlier studies [ G.E.Kodzhaspirov, A.I.Rudskoy, V.V.Rybin Effect of thermomechanical processing on structure and corrosion-mechanical properties of AISI 321 steel. 2010, Advanced Materials Research Vols. 89-91 (2010) pp. 769-772.et.al.]

Reviewer 4 Report
Overall, even though this manuscript seems to be interesting, several issues need to be addressed as follows:
1) there is no title provided for table 1.
2) more characterizations are needed to provide such as using chemical analysis to come up with such phases presented in the manuscript.
3) it is necessary to address statistics in the results.
4) introductions need to be more thoroughly reviewed.
5) in the materials and methods section, the authors indicated the mechanical properties. it is important to specifically address what they exactly are.
Author Response
Point 1:. The title provided for Table 1 has been added.
Point 2: Chemical analysis of carbide precipitates has been carried out in accordance with technique [22].
Point 3: For each mode, from 3 to 5 samples per point were used.
Point 4: Introduction has been corrected in accordance with Reviewer’s recommendations.
Point 5: In Table 1, the estimated characteristics of mechanical properties were clarified: Yield Strength (YS) and Elongation (A).

Round 2
Reviewer 1 Report
The logic of the manuscript can be improved. Some pattern problems exist. Please correct them.
The English can be polished. Some explanation should be added to highlight the core content.
Author Response
In accordance with the Reviewer's recommendation, the logic of the manuscript has been corrected. All corrections are in red.
Reviewer 3 Report
Accepted.
Author Response
In accordance with the Reviewer's recommendation English has been corrected.
All corrections are in red.

Reviewer 4 Report
-
Author Response
In accordance with the Reviewer's recommendation, English has been corrected.
All corrections are in red.
